# Toxicological Comparison of Pesticide Active Substances Approved for Conventional vs. Organic Agriculture in Europe

**DOI:** 10.3390/toxics10120753

**Published:** 2022-12-02

**Authors:** Helmut Burtscher-Schaden, Thomas Durstberger, Johann G. Zaller

**Affiliations:** 1Umweltforschungsinstitut & Umweltorganisation Global 2000 (Friends of the Earth Austria), Neustiftgasse 36, 1070 Vienna, Austria; 2Department of Integrative Biology and Biodiversity Research, Institute of Zoology, University of Natural Resources and Life Sciences Vienna (BOKU), Gregor Mendel Straße 33, 1180 Vienna, Austria

**Keywords:** agrochemicals, farming, synthetic pesticides, natural pesticides, environmental risk assessment, organic farming, pesticide reduction, farm to fork strategy

## Abstract

There is much debate about whether the (mostly synthetic) pesticide active substances (AS) in conventional agriculture have different non-target effects than the natural AS in organic agriculture. We evaluated the official EU pesticide database to compare 256 AS that may only be used on conventional farmland with 134 AS that are permitted on organic farmland. As a benchmark, we used (i) the hazard classifications of the Globally Harmonized System (GHS), and (ii) the dietary and occupational health-based guidance values, which were established in the authorization procedure. Our comparison showed that 55% of the AS used only in conventional agriculture contained health or environmental hazard statements, but only 3% did of the AS authorized for organic agriculture. Warnings about possible harm to the unborn child, suspected carcinogenicity, or acute lethal effects were found in 16% of the AS used in conventional agriculture, but none were found in organic agriculture. Furthermore, the establishment of health-based guidance values for dietary and non-dietary exposures were relevant by the European authorities for 93% of conventional AS, but only for 7% of organic AS. We, therefore, encourage policies and strategies to reduce the use and risk of pesticides, and to strengthen organic farming in order to protect biodiversity and maintain food security.

## 1. Introduction

The global use of pesticides has increased significantly in recent decades, impacting human health and the environment, and contaminating sites even far from the areas of application [1,2,3]. The majority of active substances (AS) of pesticides used in both conventional and organic agriculture not only affect target organisms but have side effects on non-target organisms, including humans [4]. According to the globally adopted but voluntary concept of “good agricultural practice”, the maintenance of high biodiversity and the promotion of beneficial organisms should be the basis of plant protection and, consequently, the use of pesticides should be kept to a minimum in order to achieve environmentally friendly agriculture, and to protect human health from the inappropriate use of agrochemicals [5].

Indeed, farmers have relied on crop rotation, mechanical plant protection measures, robust varieties, and biodiversity-enhancing farming practices to protect crops from pests and diseases since the dawn of agriculture [6]. However, the use of naturally occurring insecticides, such as pyrethrins from *chrysanthemum* plants or fungicides based on mineral copper or sulfur compounds, has also been documented since medieval agriculture [7]. Some of these naturally occurring AS are still used, especially in organic farming. However, during the 20th century, synthetic pesticides have become increasingly dominant in conventional agriculture, largely replacing naturally occurring AS and non-chemical crop protection measures [8]. Most synthetic AS act through a targeted interaction with biochemical processes that are important for the survival of populations of pests and disease organisms [9]. Unlike these pesticidal AS, microorganisms and pheromones, as well as the vast majority of plant extracts and mineral AS permitted in organic agriculture, exert their effects via a non-specific chemical, physicochemical, or physical mode of action, or by deterring or confusing pests [10].

Even the first generation of synthetic pesticides exhibited a significantly higher toxicity to their target organisms than most naturally occurring AS. As new classes of synthetic pesticides were developed, this toxicity steadily increased. On the one hand, this led to much lower hectare application rates [11], but on the other hand this led to an increase in toxicity to non-target organisms as well. For most modern pyrethroid or neonicotinoid insecticides, the LD_50_ for honeybees is three to four orders of magnitude lower than that of first general insecticides such as DDT, while sublethal effects occur at even lower concentrations, and affect colony survival [12].

With the increase of pesticide use in recent decades, pressure on ecosystems has increased worldwide [2,3]. Active substances used in organic agriculture have also been shown to affect non-target organisms: for instance, copper on soil biota [13,14], sulfur on predatory mites [15], or spinosad, an insecticide based on the bacterium *Saccharopolyspora spinosa*, being toxic to honeybees and water organisms [16]. Neonicotinoid insecticides used in conventional agriculture, but not allowed in organic farming, have been linked to detrimental effects on insect biodiversity [12,17], soil organisms [18,19], and were found in carnivorous and insectivorous birds [20]. Additionally, residues of synthetic pesticides can affect sensitive crops [21] and contaminate organically farmed soils [22,23], and even nature conservation areas [24,25,26].

Intense pesticide use contributes to the global decline in biodiversity [27,28,29] and the overshoot of planetary boundaries [30]. This loss of biodiversity ultimately undermines the foundations of agricultural production, and thus, “seriously threatens the future of our food, livelihoods, health and environment” [31]. Against this backdrop, the Commission of the European Union (EU) put forward the European Green Deal with its Farm to Fork and Biodiversity strategies, which aim, among other aspects, to halve the use and risk of chemical pesticides in the EU by 2030, and to expand organic farming to 25% of EU farmland [32]. These targets made the issue of pesticide use in agriculture a topic of political and societal debate [33]. In a statement to the EU Commission, Europe’s pesticide industry argued that an increase in organic agriculture would lead to an “increased overall volume of pesticide use in Europe”, because the hectare application rates of naturally occurring pesticides would be much higher than those of synthetic pesticides [34]. This argument is noteworthy because, at least in Germany, it is estimated that pesticides are applied on only about 5–10% of the organic agricultural land [35,36]. The warning of an ecological trade-off due to an increase in organic agriculture would, therefore, only make sense if the AS permitted in organic farming were significantly more hazardous than the chemical-synthetic AS used in conventional farming.

Assessment of the intrinsic properties of pesticide AS can give rise to scientific controversy [37,38]. Evaluating the potential toxic burden of pesticides in the field is even more difficult, due to numerous interactions with other factors [29,39]. However, in Europe, all authorized pesticide AS—whether synthetic or natural-based—must meet the same authorization criteria set out in the EU Pesticides Regulation (EC) No 1107/2009 [40]. To ensure this, applicants are required to submit standardized in vitro and in vivo tests in which physical and toxicological properties of the AS are investigated. Based on the test results, health-based guidance values, such as the Acceptable Daily Intake (ADI), Acceptable Operator Exposure Level (AOEL), and Acute Reference Dose (ARfD), are established by the regulatory authorities where relevant [41] and hazard classifications under the Globally Harmonized System of Classification and Labelling of Chemicals GHS [42] are made where appropriate. An approach for a comparative assessment of the potential toxic burden of pesticide AS may, therefore, be based on these health-based guidance values and GHS classifications established under the EU pesticide authorization procedure, as has been done in previous studies [43,44].

The objective of this study was to compare the potential toxicological hazards to humans and the environment of pesticide AS approved only for conventional agriculture versus those AS approved for use in organic agriculture. Our results will help policy makers in their quest for more sustainable agriculture in Europe.

## 2. Materials and Methods

### 2.1. Database Used for This Comparison

We accessed the EU pesticide database (https://ec.europa.eu/food/plants/pesticides/eu-pesticides-database_en) (accessed on 13 October 2022) to identify 450 pesticide AS currently authorized in the EU, after eliminating 6 database entries as duplicates [45].

Of these 450 AS, 256 AS may be applied only on conventionally farmed land, henceforth referred to as ConvAS. This number that resulted from the total of approved AS, subtracting those 10 AS which may only be used for post-harvest treatment (1-methylcyclopropene, 1,4-dimethylnaphthalene, 2-phenylphenol, aluminium sulphate, benzoic acid, carvone, phosphane, pirimiphos-methyl, sodium silver thiosulphate, and sulfuryl fluoride), according to regulation 540/2011 [46], and subtracting those AS that are permitted in organic farming.

According to Annex 1 of regulation 2021/1165 [47], 185 AS were approved for use in organic agriculture, herein referred to as OrgAS. Of these, 47 AS are used in traps only (pheromones and other semiochemicals, and the three insecticidal AS diammonium phosphate, deltamethrin, and lambda-cyhalothrin), 2 AS may only be used as storage gases only (CO_2_ and ethylene), and 2 AS may only be used for post-harvest treatment (clove oil, spear mint oil). Thus, for this comparison 134 OrgAS were considered. All OrgAS can also be applied in conventional agriculture, but not vice versa.

### 2.2. Comparison of Potential Risks for Human Health and Aquatic Toxicity

All AS were assessed regarding the following:their classification as basic substance, low-risk AS, candidate for substitution, and substances that do not fall into any of these groups [46];their health-based guidance values: Acceptable Daily Intake ADI, Acute Reference Dose ARfD, and the Acceptable Operator Exposure Level AOEL [40]; andtheir hazard classifications under the Globally Harmonized System GHS on classification, labelling, and packaging of substances and mixtures [42].

The allocation of the substances, with regard to their origin as chemically synthesized or naturally occurring inorganic or organic substances, as well as microorganisms, was based on information from the Pesticide Properties Database of the University of Hertfordshire PPDB [48]. Wherever it seemed appropriate, this information was checked and supplemented using the scientific literature.

The allocation of the AS into the groups insecticides, fungicides, herbicides, molluscicides, plant growth regulators, and “other AS”, was taken from the EU Pesticides Database [49]. For those AS not listed in the EU Pesticides Database, (e.g., basic substances), the PPDB [48], or the registration documents of the AS were consulted (details are shown in Appendix A).

A basic substance is defined in the plant protection product (PPP) Regulation (EC) No 1107/2009 as an AS that is not predominantly used as a plant protection product, but which may be of value for plant protection, and for which the economic interest in applying for approval may be limited [40,50]. Basic substances must not be substances of concern, and must not have an inherent capacity to cause endocrine disrupting, neurotoxic, or immunotoxic effects [51]. In particular, AS that meet the criteria of a “food”, as defined in article 2 of regulation (EC) 178/2002, are classified as basic substances [52]. All basic substances have to meet the approval requirements of the PPP regulation, but are subject to a simplified authorization procedure.

A low-risk AS is expected to pose only a low risk to human and animal health, and to the environment [40,53]. Low-risk AS must not be carcinogenic, mutagenic, toxic to reproduction, sensitizing chemicals, very toxic or toxic, explosive, corrosive, persistent, and have a bioconcentration factor >100. Active substances of low risk must not be deemed to be endocrine disruptors, or have neurotoxic or immunotoxic effects.

A candidate for substitution is defined as a pesticide AS that meets one or more of the following criteria: it is or is to be classified, in accordance with the provisions of Regulation (EC) 1272/2008, as carcinogenic or as toxic for reproduction (category 1A or 1B), or it is or it is considered to have endocrine-disrupting properties that may cause adverse effects in humans [40,54,55]; it meets two of the criteria to be considered as persistent, bioaccumulative, and toxic (PBT criteria); it gives reasons for concern regarding developmental neurotoxic or immunotoxic effects; it has a high potential of risk to groundwater; or it contains a significant proportion of non-active isomers.

The health-based guidance values, ADI as an estimate for safe lifetime dietary exposures, ARfD as an estimate for safe acute, one-meal or one-day exposures; and AOEL, as an estimate for safe non-dietary exposures, are established, where relevant, by the European Food Safety Authority EFSA in the course of the authorization procedure of AS [40].

Finally, hazard statements on health and environmental hazards were established on the basis of the Classification, Labelling, and Packaging (CLP) Regulation ((EC) No 1272/2008 [54]. These hazard statements are assigned by the European Chemicals Agency (EChA) for pesticide AS that meet the criteria for classification in accordance with this regulation, based on the GHS classification [42]. The GHS hazard statements are assigned to a hazard classification that describes the nature of the hazards of a pesticide AS. As part of the authorization procedure, manufacturers carry out animal studies in accordance with OECD guidelines. These include studies to determine acute oral and dermal toxicity as well as toxicity by inhalation, skin and eye irritation, damage to organs, and carcinogenicity and reproductive toxicity. Hazard statements on environmental hazards refer to hazards to aquatic organisms, combined with environmental fate parameters. Appendix A lists all GHS codes and hazard descriptions considered in this study.

### 2.3. Statistical Analyses

We performed Chi^2^ independence tests to determine whether frequencies of certain hazard categories differed between ConvAS and OrgAS. As an example, for the Acute toxicity-swallowed classification, we tested whether the number of assigned AIs to categories H300 (fatal if swallowed), H301 (toxic if swallowed), and H302 (harmful if swallowed) was significantly different between ConvAS and OrgAS. No test could be performed for hazard classifications with only one category. All tests were performed using SPSS version 24 (IBM Incorporation, Armonk, NY, USA).

## 3. Results

### 3.1. Comparison of Pesticide Categories

Of the 256 AS that may only be used in conventional agriculture (ConvAS), 35.5% were herbicides, 32.4% fungicides, 17.6% insecticides, 8.6% comprised plant growth regulators, and 5.1% fell into the category “other AS”, such as rodenticides, repellents, or other substances whose effect could not be clearly assigned to any of the above categories (Figure 1). Of the 134 AS that may be used in organic agriculture (OrgAS), 45.5% were fungicides, 32.1% were insecticides, and 20.9% were assigned to other categories (Figure 1). Pelargonic acid was assigned to herbicides according to its code in the database; however, herbicides are not permitted in organic farming in Europe. Plant growth regulators are also not permitted in organic farming in Europe.

### 3.2. Comparison Based on Substance Origin

The vast majority (87.9%) of the 256 ConvAS consist of synthesized organic derivates of the petroleum chemistry, 7.0% of natural organic origins, 2.7% were products of inorganic syntheses (mainly phosphides), and 2.3% were of natural inorganic origins (Figure 2). It should be mentioned that four of these natural convAS (aqueous extracts from the germinated seeds of sweet *Lupinus albus*, ABE-IT 56, talcum E553B, ferric pyrophosphate) are currently (as of November 2022) in a process of approval for use in organic farming, and are expected to be approved in the near future; others may follow later.

All 134 OrgAS were natural or naturally-derived substances, as required by the EU Organic Regulation (EU) 2018/848 [56]. Of these, 56.0% consisted of microorganisms such as bacteria, viruses, or fungi; 32.1% were of natural organic origins (e.g., consisting of essential oils and other plant extracts with fungicidal, insecticidal, or deterrent activity; substances of animal origins, such as sheep fat as a repellent); and 11.9% were of natural inorganic origins (e.g., minerals, salts, and elemental substances based on copper, sulfur, iron, silicon, phosphorus, sodium, and potassium) (Figure 1).

A full list of all individual active substances is provided in Appendix A.

### 3.3. Comparison of Regulatory Risk Ratings

Of the 256 ConvAS, 18.7% were classified as candidates for substitution, 2.3% were low-risk AS, 0.8% were basic substances, and 78.2% were not classified (Figure 3).

Of the 134 OrgAS, 22.4% were low-risk AS, 14.2% were basic substances, 3.7% were candidates for substitution (all consisting of copper compounds), and 59.7% were not classified. More detailed information is provided in Appendix A.

### 3.4. Comparison Based on Health-Based Guidance Values ADI, ARfD, AOEL

Health-based guidance values for dietary and non-dietary exposures were set by the EFSA for 93.0% of all ConvAS: an ADI was thereby established for 93%, an ARfD for 61.7%, and an AOEL for 93.0%. Of the OrgAS, only 5.2% AS had an ADI, 2.2% an ARfD, and 6.0% AS an AOEL. For the remaining 93.3%, the establishment of health-based guidance values was not deemed relevant. Within the OrgAS, the insecticides spinosad, pyrethrins, and azadirachtin, and the fungicide thymol, showed the lowest acceptable dietary and non-dietary exposure levels, which were in the range between 0.1 and 0.01 mg kg^−1^ of body weight. The lowest acceptable exposure levels within the ConvAS were two orders of magnitude lower (between 0.001 and 0.0001 mg kg^−1^ of body weight), and concerned five synthetic herbicides, tembotrione, sulcotrione, fluometuron, metam (also a nematicide, insecticide, and fungicide), and diclofop, and two insecticides, emamectin and oxamyl (Figure 4).

### 3.5. Comparison of Globally Harmonized Hazard Statements

Of the ConvAS, 54.7% (140 AS) carried hazard statements; however, only 3.0% (4 AS) of OrgAS comprised the two insecticides pyrethrins and spinosad, the fungicide sulfur, and the basic substance hydrogen peroxide (Figure 5). The maximum number of hazard statements assigned to a single AS was nine for ConvAS, and five for OrgAS.

Regarding individual hazard statements, ConvAS were significantly more hazardous than OrgAS (Figure 6). Regarding acute toxicity if swallowed, 24.6% (63 AS) of ConvAS were either harmful (H302), toxic (H301), or fatal (H300) if swallowed compared to 1.5% (2 AS) of OrgAS. Regarding acute toxicity on skin contact, 3.5% (9 AS) of ConvAS were considered harmful (H312), toxic (H311), or fatal (H310) in contact with skin, while in OrgAS, 0.7% (1 AS) were in this category. Regarding acute toxicity if inhaled, 11.5% (30 AS) of ConvAS were either harmful (H332), toxic (H331), or fatal (H330) if inhaled, while 1.5% (2 AS) of OrgAS were in this category. Regarding skin damage, 25.0% (64 AS) of ConvAS may cause allergic skin reaction (H317), cause skin irritation (H315) or severe skin burns (H314), while 1.5% (2 AS) of OrgAS fell in this category. Regarding eye damage, 10.9% (28 AS) of ConvAS caused eye irritation (H320), serious eye irritation (H319), or serious eye damage (H318, H314), while 0.7% (1 AS) of OrgAS fell in this category. Of the ConvAS, 6.6% (17 AS) cause (H370) or may cause (H371) damage to organs, 7.8% (20 AS) may (H360, H360D) damage or were suspected of damaging the unborn child (H361d, H361fd), and 6.6% (17 AS) were suspected of causing cancer (H351), while none of these hazard categories were assigned to OrgAS (Figure 6).

Of the 134 OrgAS, only 3.0% (4 AS) carried hazard statements: the two insecticidal AS spinosad derived from the actinobacterium *Saccharopolyspora*, and pyrethrin derived from the plant species *Chrysanthemum cinerariifolium*, as well as the fungicidal compounds hydrogen peroxide and sulfur. Pyrethrins were harmful if swallowed (H302) or inhaled (H332), and harmful in contact with skin (H312). In addition, pyrethrins and spinosad must be labelled as very toxic to aquatic organisms (H400) and very toxic to aquatic organisms, with long-lasting effects (H410). Elemental sulfur causes skin irritation (H315), while hydrogen peroxide causes severe skin burns and eye damage (H314), and is also harmful if swallowed (H302) or inhaled (H332).

Regarding acute aquatic toxicity 39.8% (102 AS) of ConvAS were classified as very toxic to aquatic life, but only 1.5% (2 AS) of OrgAS, namely the two insecticides pyrethrins and spinosad (Figure 7). Regarding chronic aquatic toxicity, 49.6% (127 AS) of ConvAS were harmful, toxic, or very toxic to aquatic life with long-lasting effects, while only 1.5% (2 AS, pyrethrins and spinosad) of OrgAS were classified as very toxic to aquatic life with long-lasting effects. For more detailed information, see Appendix A.

All differences described above were highly significant using the chi^2^ test (see Appendix A for test results). These differences remained even when all low-risk AS, basic substances, and microorganisms were excluded from the comparison.

## 4. Discussion

To our knowledge, this is one of the first comprehensive evaluations of the potential toxicity to humans and the aquatic environment of pesticide AS approved for use in conventional agriculture (ConvAS) in the EU, compared to those approved for use in organic agriculture (OrgAS), based on the risk and hazard classifications considered in the EU pesticide authorization process. Overall, we found that ConvAS have a significantly higher potential hazard to humans and the environment than OrgAS, in all categories considered. Since our assessment was based on AS only, the inclusion of co-formulants would most likely result in an even greater hazard potential from ConvAS [57,58,59].

The health-based guidance values and the GHS hazard statements that build the basis of our analysis cover a wide range of health-related regulatory studies submitted for the EU approval process. However, environmental effects in the GHS hazard classes are limited to the assessment of acute and chronic toxicity to aquatic organisms only, and include toxicity studies on fish, crustaceans, daphnia, and algae, as well as degradation and bioaccumulation of AS. Effects on pollinators, birds, and earthworms, as well as on groundwater, which also contribute to the data requirements of the EU approval procedure, are currently not considered in the GHS hazard classifications.

Our assessment has shown that 55% (140 AS) of ConvAS currently approved in the EU carry health or environmental hazard statements, while only 3% (4 AS) of OrgAS do. Hazard statements warning of harm to the unborn child, cancer, or lethal effects from inhalation, oral, or dermal intake, were found in 16.0% of ConvAS, while none of the OrgAS were associated with these hazard classes. In addition, the establishment of health-based guidance values for dietary exposure (ADI, ARfD) or occupational exposure (AOEL) was considered relevant by the EFSA for 93.0% of ConvAS (238 AS), but only for 6.7% (9 AS) of OrgAS.

One explanation for this significant difference in toxicity lies in the nature and origin of the respective pesticide AS. Synthetic ConvAS are selected in laboratory screening programs to identify substances with particularly high toxicity to target organisms. For example, the toxicity of synthetic insecticides approved in the USA has approximately doubled in 10 years, although application rates have been halved [60]. Similarly, the toxicity of herbicides applied in Austria to honeybees, earthworms, or birds increased by more than 400%, while their use, as measured in kg of AS, decreased by 24% [44].

In comparison, most of the OrgAS included in our assessment posed a much lower risk, simply because they are approved as low-risk AS (e.g., iron phosphate, baking powder, yeast extracts, or microorganisms), as basic substances (e.g., onion oil, washing soda, vinegar, or milk) or as microorganisms [45]. However, even if all low-risk AS, basic substances, and microorganisms are excluded from the comparative assessment, the differences in the proportion of hazard classifications remain statistically significant, as of the remaining 34 OrgAS, more than 90% still carry no hazard statements.

Another explanation for this fundamental difference between ConvAS and OrgAS lies in their different modes of action. Almost all chemically synthesized ConvAS exert their effect by influencing biochemical processes in the respective target organisms or non-target organisms, in the case of undesirable side effects [9]. In this context, most synthetic AS act as “single-site” inhibitors of enzymes or transmembrane receptors that are essential for cell metabolism and signaling.

In the case of OrgAS, to our knowledge, a single-site mode of action was found only in three insecticides: two plant secondary compounds, azadirachtin and pyrethrins, and the bacterial agent spinosad [10]. While azadirachtin inhibits hormonally induced molting of insect larvae, both pyrethrins and spinosad inhibit the transmission of nerve impulses. Interestingly, these three natural insecticides alone account for seven of the eleven health and environmental hazard statements, and about one-third of the health-based guidance values of all 134 OrgAS evaluated.

All other OrgAS in the EU generally have a multi-site mode of action, or act in other ways by driving away pests, or by enhancing the plant’s defenses, which is the main reason why the development of resistance is rarely observed with OrgAS in contrast to ConvAS [61]. OrgAS such as copper or sulfur affect cellular processes in fungi at different levels simultaneously [62]. Other OrgAS, such as vinegar or soap, act in a physicochemical way by damaging the cell membrane. Baking soda (potassium hydrogen carbonate) or slaked lime (calcium hydroxide) alter the pH and desiccate the target organism, while plant oils form a physical barrier between the plant and insect pests [10]. Substances such as garlic extract or quartz sand act as repellents via odor or taste.

As a result of the differences in the toxicity and mode of action described above, the vast majority of OrgAS (with the exception of microorganisms and pheromones) require hectare application rates that are one to three orders of magnitude higher than typical ConvAS (i.e., kg ha^−1^ for OrgAS vs. g ha^−1^ for ConvAS). Given this wide range in hectare application rates and toxicity, it is self-explanatory that the risk associated with the use of different pesticide AS cannot be assessed by simply adding up the quantities used in kilograms. Pesticide risk indicators that do this, and do not consider toxicity or hectare application rates of each AS, can subsequently lead to grossly distorted results. They systematically underestimate the risk of (mostly synthetic) pesticides that have low per-hectare application rates due to their high toxicity, compared to (usually natural) pesticides that have high per-hectare application rates and low toxicity [11,44]. This misconception applies in particular to the Harmonized Risk Indicator 1 (HRI 1) [63], currently proposed to monitor the pesticide reduction targets of the Farm to Fork Strategy [32,64].

Our comparative assessment demonstrates that the warnings of an ecological trade-off due to an increase in organic agriculture, as expressed by Europe’s pesticide industry [34], are not supported by the hazard classifications and health-based guidance values assigned by the EU authorities, since they show OrgAS to have a significantly lower toxicity than ConvAS. However, one may argue that the benefit of lower toxicity could be outweighed by the higher hectare application rates of OrgAS, since exposure considerations are outside the scope of hazard assessments, and are, therefore, also outside of this evaluation. However, again, the available scientific literature provides little evidence of a relevant contribution of OrgAS to the adverse effects of pesticides on soils, aquatic ecosystems, pollinating insects, and other non-target organisms [3,65,66], or on pesticide-related acute and chronic health effects in humans [67,68,69]—at least at the levels at which OrgAS are currently used.

An important reason for the observed inconspicuousness of OrgAS with regard to negative health and environmental impacts could be, in addition to their significantly lower hazard potential, their integration into the planetary material cycles [70]. This is particularly evident for plant-, animal-, and microorganism-derived substances, which are integrated into the carbon and water cycles and—depending on their chemical composition—also into the nitrogen, phosphorus, or sulfur cycles. Microbial and plant-based OrgAS, or even the toxic natural insecticides spinosad, azadirachtin, and pyrethrins, are usually biodegraded quite quickly [71].

Inorganic OrgAS are embedded in natural cycles too. They are subject to chemical transformations and weathering processes, in the course of which their biological activity tends to be reduced. Sulfur, for example, occurs naturally in organic and inorganic forms in a proportion of 0.2 to 5.0 g kg^−1^ in soil. When elemental sulfur is applied as a fungicide, it is usually oxidized in the soil, and is thus mineralized to plant-available sulfate, with an average half-life in active soils of less than one day [71]. Calcium hydroxide, which can be used as a fungicide in fruit growing, is rapidly converted to calcium carbonate, which is one of the most common deposits in the earth’s crust. Potassium hydrogen carbonate also occurs naturally in soil and dissociates in aqueous solution into potassium ions (K^+^), which are a natural component of soil and water, and hydrogen carbonate ions (HCO_3_^−^) are part of the carbon cycle in equilibrium with CO_2_ derived from plant roots and the atmosphere. Naturally occurring background levels of bioavailable potassium are well above the maximum levels that could be released into soil and water during fungicide treatment with potassium hydrogen carbonate [10]. Potassium, calcium, and sulfur are essential nutrients or micronutrients for plants. This is also true for copper, an important fungicidal AS for both organic and conventional agriculture, which occurs naturally in the soil in various forms. Biologically active free copper ions, such as those contained in copper-containing fungicides, are largely adsorbed by the clay–humus complex when they enter the soil, with the proportion of biologically active copper in the soil being less than 0.3% of the total detectable copper [72]. Consequently, a meta-analysis on the ecotoxicity of copper shows that the hectare application rate of 4 kg Cu ha^−1^ year^−1^ currently authorized for viticulture in Europe does not significantly alter the biological quality and functions of the soil [73].

Five mineral copper compounds that are currently authorized in the EU (copper oxychloride, copper oxide, copper hydroxide, copper sulphate, and Bordeaux mixture) were the only OrgAS classified as candidates for substitution in Europe [74]. Copper fulfills the formal criterion of a candidate for substitution, due to its toxicity to aquatic organisms and its persistence. However, it is controversial whether the concept of persistence, which was developed to describe the resistance of certain chemically synthesized organic compounds (so-called POPs; persistent organic pollutants) to chemical, physical, and biological degradation, should also be applied to metallic or mineral compounds. Indeed, the GHS classification is quite clear in this respect: “For inorganic compounds and metals, the concept of degradability as applied to organic compounds has limited or no meaning” [42]. The EFSA has acknowledged in its assessment report on copper that the available guidance in the area of environmental risk assessment does not specifically cover metal compounds, and that the assessment of copper was, therefore, carried out in the light of currently available methods [75]. In the meantime, the EFSA has published a “statement on environmental exposure and risk assessment for transition metals”, based on which copper has been currently reassessed in the EU [76].

In view of the climate and biodiversity crisis, low-input farming has increasingly become the beacon of hope for a climate- and biodiversity-friendly agriculture of the future. This is also reflected in the goals of the European Green Deal, which, with its Farm to Fork and Biodiversity strategies, aims, among other things, to reduce fertilizer use by 20% and antibiotic and pesticide use by 50% by 2030, and to expand organic farming from the current 8% to 25% of EU arable land [32].

Plant protection in organic farming differs conceptually and, above all, in practice from conventional farming: preventive measures form the basis, and all practices are regularly monitored [77]. Organic farming rules allow for the use of pesticide AS only as a last resort, and include provisions for increasing biodiversity, crop rotation, soil conservation and health, and area-based livestock [77]. As a result, only a small proportion of organic areas is treated with pesticides, often ranging from 5% to a maximum of 10% [36]. Furthermore, organic agriculture benefits overall biodiversity [35,78]. Even though our analysis showed that OrgAS have a significantly lower hazard potential than ConvAS, all efforts to further reduce dependence on OrgAS in these areas are welcome. This can be achieved by intensifying research on agroecological cultivation and plant protection methods, and by developing robust, fungus-resistant varieties.

## 5. Conclusions

Our assessment shows that pesticide AS, approved for use in conventional and integrated agriculture, are clearly more hazardous to humans and the environment than the naturally occurring AS approved for use in organic agriculture. Claims from the pesticide industry, in which the expansion of organic agriculture envisaged in the European Farm to Fork Strategy could lead to ecological trade-offs due to an increase in the use of natural pesticides, are clearly not supported by the results of our analysis. We, therefore, encourage any political strategy that aims to reduce the use and risk of chemical pesticides while increasing the acreage of organic agriculture. This will help to reduce hazards to human health, the environment, and biodiversity, and thus preserve the ecosystem services that are essential to maintaining food security.

## Figures and Tables

**Figure 1 toxics-10-00753-f001:**
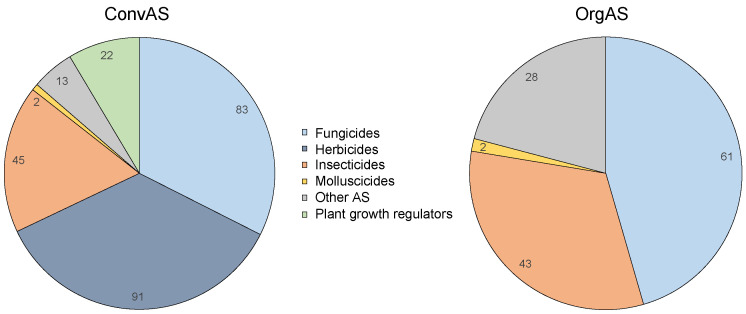
Assignment of active substances (AS) approved in conventional (ConvAS, *n* = 256) or organic agriculture (OrgAS, *n* = 134) to pesticide categories. The numbers in the sectors indicate the number of AS.

**Figure 2 toxics-10-00753-f002:**
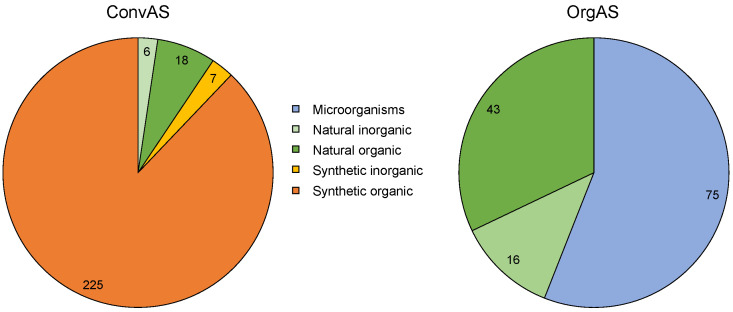
Origin of active substances (AS) approved in conventional (ConvAS, *n* = 256) or organic agriculture (OrgAS, *n* = 134) in the European Union. Numbers represent the number of AS of a particular origin class. The numbers in the sectors indicate the number of AS.

**Figure 3 toxics-10-00753-f003:**
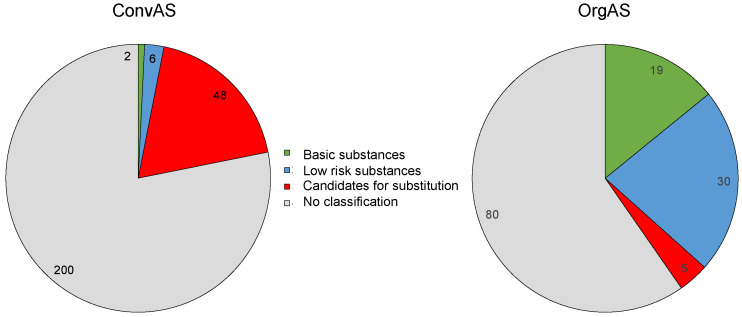
Comparison of basic risk classifications of active substances (AS) approved for use in conventional (ConvAS; *n* = 256) and organic agriculture (OrgAS, *n* = 134).

**Figure 4 toxics-10-00753-f004:**
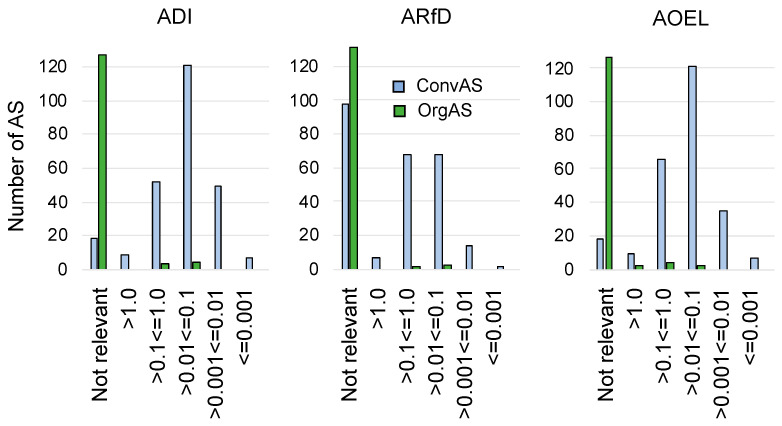
Comparison of ADI (Acceptable Daily Intake), ARfD (Acute Reference Dose), and AOEL (Acceptable Operator Exposure Level) of active substances (AS) approved for use in conventional (ConvAS, *n* = 256) and organic agriculture (OrgAS, *n* = 134).

**Figure 5 toxics-10-00753-f005:**
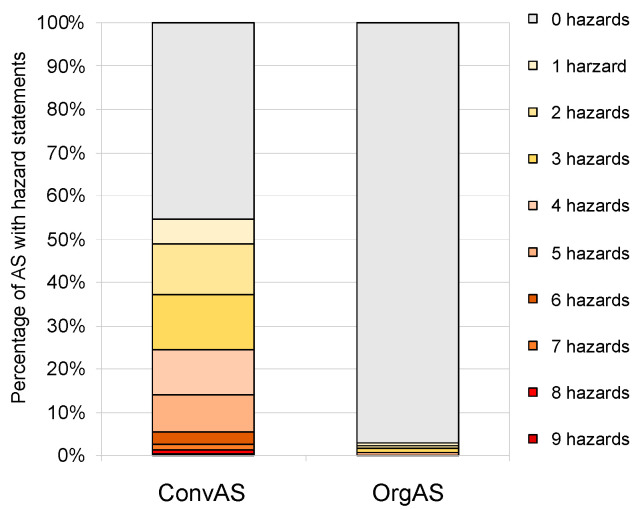
Comparison of the relative numbers of human toxicity hazard statements of active substances (AS) approved for use in conventional (ConvAS, *n* = 256) and organic agriculture (OrgAS, *n* = 134).

**Figure 6 toxics-10-00753-f006:**
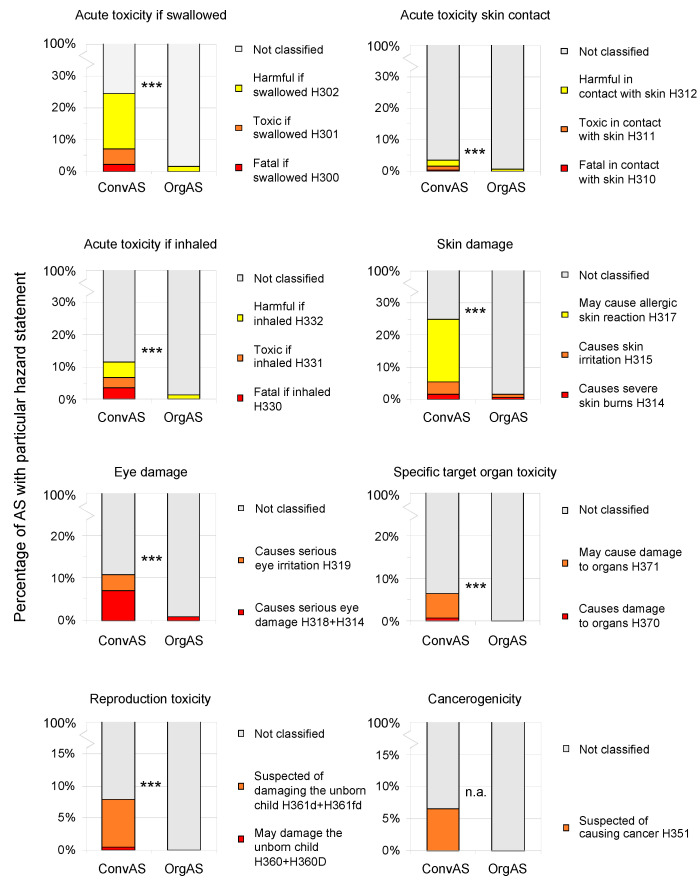
Comparison of human health-related hazard statements of active substances (AS) approved for use in conventional (ConvAS, *n* = 256) and organic agriculture (OrgAS, *n* = 134). Results of chi^2^ tests denoted with asterisks: *** *p* < 0.001, n.a. a chi^2^ test was not applicable because of too few categories.

**Figure 7 toxics-10-00753-f007:**
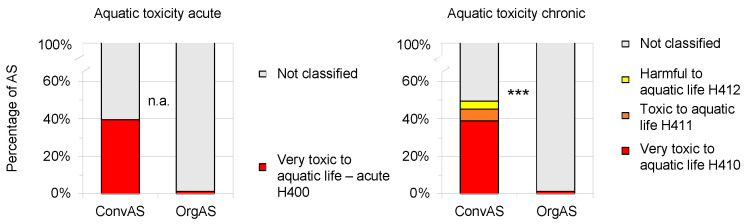
Comparison of acute and chronic aquatic toxicity of active substances (AS) approved for use in conventional (ConvAS, *n* = 256) and organic agriculture (OrgAS, *n* = 134). Results of chi^2^ tests denoted with asterisks: *** *p* < 0.001; n.a. a chi^2^ test was not applicable because of too few categories.

## Data Availability

All data are provided in the Appendix A of this paper.

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
