# Peer review of "Toxicological Comparison of Pesticide Active Substances Approved for Conventional vs. Organic Agriculture in Europe"

_toxics, 2022, doi:10.3390/toxics10120753_

Round 1

Reviewer 1 Report

Current manuscript entitled “Toxicological Comparison of Pesticide Active Substances Approved for Conventional vs. Organic Agriculture in Europe” by “Burtscher-Schaden et al”. evaluated the potential toxicological hazards to humans and the environment from pesticide AS approved for conventional agriculture compared to AS approved for use in organic agriculture. Obtained results can help policy makers in their quest for a more sustainable agriculture in Europe. The manuscript seems interesting and can be accepted after addressing the following comments.

1.     Conclusions section needs to be improved.

2.     Provide the full form of ADI, ARfD and AOEL in the abstract section.

3.     How the statistical analyses were performed? elaborate in detail.

4.     Revise the last paragraph of the introduction section.

Author Response

Dear Reviewer #1

Thank you for your appreciation of our work.

We have revised our manuscript according to your suggestions/comments and included the references you kindly suggested.

We hope that the revised manuscript can now be accepted for publication.

Yours sincerely,

Johann G. Zaller, on behalf of all authors

Reviewer 2 Report

General: Few typos in % (may be ED for endocrine disruptors too as abrev.)

Of course nice and important work.

Some ref for specific AS (basic, low-risk, CfS) mentioned some precise references could be added (suggested)

Take care of clayed charcoal, p5 "may be validated laer in OP" (not voted nov. 2022)

May be insist on the fact that PBT criteria in case of 2 for CfS (i.e. copper compounds) are not any more written in pesticide database for CfS.

Author Response

Dear Reviewer #2,

Thank you for your appreciation of our work.

We have revised our manuscript according to your suggestions/comments.

We hope that the revised manuscript can now be accepted for publication.

Yours sincerely,

Johann G. Zaller, on behalf of all authors

Round 2

Reviewer 1 Report

The revised manuscript can be accepted for publication

Reviewer 2 Report

Good improvements resulting from both evaluators questions and request.

More details, references calculations and explanations.

This work has to be published!